# Possible Repositioning of an Oral Anti-Osteoporotic Drug, Ipriflavone, for Treatment of Inflammatory Arthritis via Inhibitory Activity of KIAA1199, a Novel Potent Hyaluronidase

**DOI:** 10.3390/ijms23084089

**Published:** 2022-04-07

**Authors:** Hiroshi Koike, Yoshihiro Nishida, Tamayuki Shinomura, Bisei Ohkawara, Kinji Ohno, Lisheng Zhuo, Koji Kimata, Takahiro Ushida, Shiro Imagama

**Affiliations:** 1Department of Orthopedic Surgery, Nagoya University Graduate School of Medicine, 65 Tsurumai, Showa, Nagoya 466-8550, Japan; hiroshikoike@med.nagoya-u.ac.jp (H.K.); takusan_yjp@yahoo.co.jp (L.Z.); imagama@med.nagoya-u.ac.jp (S.I.); 2Department of Rehabilitation Medicine, Nagoya University Hospital, 65 Tsurumai, Showa, Nagoya 466-8550, Japan; 3Department of Hard Tissue Engineering, Tokyo Medical and Dental University, Tokyo 113-8510, Japan; s.tamayuki@gmail.com; 4Center for Neurological Diseases and Cancer, Division of Neurogenetics, Nagoya University Graduate School of Medicine, 65 Tsurumai, Showa, Nagoya 466-8550, Japan; biseiohkawara@med.nagoya-u.ac.jp (B.O.); ohnok@med.nagoya-u.ac.jp (K.O.); 5Multidisciplinary Pain Center, Aichi Medical University, Nagakute 480-1195, Japan; kimata@aichi-med-u.ac.jp (K.K.); ushidat@aichi-med-u.ac.jp (T.U.)

**Keywords:** KIAA1199, inflammatory arthritis, hyaluronan, hyaluronidase, collagen-induced arthritis, CEMIP

## Abstract

KIAA1199 has a strong hyaluronidase activity in inflammatory arthritis. This study aimed to identify a drug that could reduce KIAA1199 activity and clarify its effects on inflammatory arthritis. Rat chondrosarcoma (RCS) cells were strongly stained with Alcian blue (AB). Its stainability was reduced in RCS cells, which were over-expressed with the KIAA1199 gene (RCS-KIAA). We screened the drugs that restore the AB stainability in RCS-KIAA. The effects of the drug were evaluated by particle exclusion assay, HA ELISA, RT-PCR, and Western blotting. We further evaluated the HA accumulation and the MMP1 and three expressions in fibroblast-like synoviocytes (FLS). In vivo, the effects of the drug on symptoms and serum concentration of HA in a collagen-induced arthritis mouse were evaluated. Ipriflavone was identified to restore AB stainability at 23%. Extracellular matrix formation was significantly increased in a dose-dependent manner (*p* = 0.006). Ipriflavone increased the HA accumulation and suppressed the MMP1 and MMP3 expression on TNF-α stimulated FLS. In vivo, Ipriflavone significantly improved the symptoms and reduced the serum concentrations of HA. Conclusions: We identified Ipriflavone, which has inhibitory effects on KIAA1199 activity. Ipriflavone may be a therapeutic candidate based on its reduction of KIAA1199 activity in inflammatory arthritis.

## 1. Introduction

In patients with inflammatory arthritis, such as osteoarthritis (OA) and rheumatoid arthritis (RA), pathophysiological changes in an involved joint cause pain, and subsequent gait disturbances, impairing the activities of daily living and quality of life (QOL). OA is a debilitating joint disease associated with substantial morbidity, including disability, reduced QOL, and financial burden [1,2]. RA is a progressive, destructive, and systemic autoimmune disease characterized by chronic synovial joint inflammation, including synovial hyperplasia, infiltration of inflammatory cells, fibrin deposition, and joint destruction [3]. Hyaluronan (HA) is a major component of the extracellular matrix (ECM) of articular cartilage and maintains the high quality of the ECM together with aggrecan in articular cartilage, giving the tissue the ability to resist compression. The prominent reduction of aggrecan in articular cartilage is a significant early event that occurs in OA [4]. Similarly, in OA, it has been reported that HA, to which many aggrecans are bound, is also reduced from cartilage in synchronization with the decrease in aggrecan [5]. HA metabolism in the synovium, as well as articular cartilage, plays an important role in the pathophysiology of OA and RA. The expression and activity of HA synthase (HAS) and HA-degrading enzyme in the synovium have been reported to change in OA and RA [6,7]. Such changes in arthritic environments cause a reduction in the molecular weight (Mw) of HA in the joint fluid [8]. Low molecular weight HA (LMW-HA) in synovial fluid via HA degradation exerts pro-inflammatory and pro-angiogenic effects [9]. Conversely, previous reports demonstrated that the serum concentration of HA in patients with OA or RA is significantly higher than that in healthy control and that the serum concentration of HA correlated with the disease severity of OA and RA [9,10,11,12].

Regarding HA metabolism in inflammatory arthritis, it has long been thought that HYAL1 and HYAL2 play a major role in HA degradation. However, due to their weak enzyme activity [13], other enzymes for HA degradation are currently considered. Yoshida et al. reported that KIAA1199, which is encoded by cell migration inducing hyaluronidase 1 gene CEMIP, has a strong hyaluronidase activity compared to the known hyaluronidase—the expression of which is upregulated in the synovium of OA or RA patients. This suggests an important role for HA degradation in the synovium of arthritis [14]. Based on the results of these previous reports, we hypothesized that suppressing the enzymatic activity of KIAA1199 might inhibit the progression of inflammatory arthritis. The purpose of this study was to search for a drug that suppresses the hyaluronidase activity of KIAA1199 by the drug repositioning method and to verify the inhibitory effect of the identified drug on arthritis in vitro and in vivo.

## 2. Results

### 2.1. Ipriflavone, a Member of the Class of Isoflavones, Were Identified as an Inducer of Alcian Blue Staining

To search for off-label effects of pre-approved drugs to restore the Alcian blue stain of KIAA1199-expressed-RCS (RCS-KIAA) cells, we first screened a library of 1186 FDA-approved drugs. We found that Ipriflavone, a member of the class of isoflavones that has long been used for osteoporosis, is an effective drug and used it in subsequent experiments. Ipriflavone is known to inhibit bone resorption, maintain bone density, and prevent osteoporosis in postmenopausal women; however, a clinical trial reported in 2001 that it was not effective in the prevention or treatment of osteoporosis. Because of the report, Ipriflavone was included in the Prestwick FDA-approved panels, but not FDA-approved, and therefore cannot be bought unless it has been prescribed by a physician in Japan.

### 2.2. Ipriflavone Restored the Alcian Blue Staining of RCS-KIAA Cells, Increased Pericellular Matrix Formation, and Prevented HA from Leaking into the Culture Medium

Ipriflavone significantly restored the Alcian blue staining of RCS cells, which is reduced by the introduction of KIAA1199 in a dose-dependent manner (Figure 1A). In the particle exclusion assay, which can evaluate pericellular matrix formation, the proportion of the pericellular matrix area was significantly increased in a dose-dependent manner by Ipriflavone treatment in RCS-KIAA cells (Figure 1B,C). The amount of HA in the cell culture medium decreased in a dose-dependent manner by Ipriflavone treatment in RCS-KIAA cells (Figure 1D).

### 2.3. Ipriflavone Inhibited the Depolymerization of HA in FLS Stimulated with TNF-α

To investigate Ipriflavone treatment on the HA of RA patients, we used RA patient-derived fibroblast-like synoviocytes (FLS). We confirmed that stimulation of TNF-α, a ligand secreted in the inflammatory region of the knee of RA patients [15], increased HA in the cultured medium and decreased it in the pericellular pool of RCS-KIAA cells (Figure 2A,B). The increment in the cultured medium and the decrement in the pericellular pool of RCS-KIAA cells tended to be rescued by Ipriflavone treatment (*p* = 0.65 in Figure 2A and *p* = 0.08 in Figure 2B). The Mw of HA in the cultured medium of TNF-α-stimulated FLS was shifted to the column with markers of higher Mw by the Ipriflavone treatment (Figure 2C).

### 2.4. Ipriflavone Suppressed the Expression of Matrix Metalloproteinases in FLS Stimulated with TNF-α

Stimulation of FLS with TNF-α increased the mRNA expression of both *MMP1* and *MMP3**p* < 0.001 and *p* < 0.001, respectively). Administration of Ipriflavone significantly suppressed these *MMP1* and *MMP3* mRNA expressions in a dose-dependent manner (*p* = 0.028 and *p* < 0.001, respectively, Figure 3A,B). Similarly, the expressions of both MMP1 and MMP3 proteins in FLS were increased by TNF-α administration (*p* < 0.001 and *p* < 0.001, respectively). Ipriflavone also decreased the protein expressions of both MMP1 and MMP3 in a dose-dependent manner (*p* = 0.06 and 0.029, respectively; Figure 3C,D).

### 2.5. Ipriflavone Suppressed Arthritis Score in Collagen-Induced Arthritis (CIA) Mice

The arthritis score based on physical examination was significantly improved from day 33 until day 41 in both the 100 mg/kg and 200 mg/kg groups, as compared with the control group (*p* < 0.05, Figure 4A). The number of infiltrating inflammatory cells in the synovium decreased in the Ipriflavone-administered group compared to the control group (Figure 4B). The histological score was significantly decreased by the administration of Ipriflavone in a dose-dependent manner (Figure 4C, *p* = 0.042).

### 2.6. Ipriflavone Suppressed the Serum Concentration of HA in CIA Mice and Reduced the Accumulation of HA in Joints

The serum concentration of HA was significantly suppressed in a dose-dependent manner by Ipriflavone administration (Figure 5A, *p* = 0.001). HA deposition in the synovium was suppressed in the Ipriflavone-administered group (Figure 5B).

## 3. Discussion

Based on the assumption that the HYAL family is the major hyaluronidase in both the cell surface and intracellular compartment, research on HA degrading activity has been focused on HYAL1 and HYAL2 [16]. However, the recent identification of new HA-degrading enzymes, KIAA1199 [14] and transmembrane 2 (TMEM2) [17], has prompted us and others to determine the mechanism of HA degradation in OA or RA patients. Recently, several reports and commentaries have focused attention on the involvement of KIAA1199 in inflammatory arthritis [5,18,19,20]. In a study of KIAA1199 in inflammatory arthritis synovium, Yoshida et al. reported that KIAA1199 has a strong hyaluronidase activity compared to the other known hyaluronidase enzymes (HYAL1, 2, and CD44). Furthermore, they reported enhanced HA degradation in synovial fibroblasts obtained from patients with OA or RA, which had increased levels of KIAA1199 expression. In addition, knockdown of KIAA1199 inhibited the degradation activity in human osteoarthritic cartilage, suggesting that KIAA1199 may play a major role in HA degradation in the synovium of arthritis [5]. Yang et al. reported that KIAA1199 expression in synovial fibroblasts obtained from RA patients was significantly higher than in healthy controls [18]. They also reported upregulation of KIAA1199 at mRNA and protein levels in synovial tissue, serum, and joint fluids of RA patients compared to healthy controls, suggesting the KIAA1199 expression as a diagnostic biomarker of RA [19].

On the other hand, the destruction of articular cartilage by protease, especially the extracellular matrix, is an important factor in the pathogenesis of OA patients. Several reports are available on KIAA1199 expression in OA cartilage. Shimizu et al. showed that KIAA1199 was mainly stained in the weakly stained area of hyaluronic acid-binding protein (HABP) in human OA cartilage and that the stainability of KIAA1199 correlates with the Mankin score, a histopathological classification of the severity of the OA lesions of cartilage [5]. They also showed that knockdown of KIAA1199 by siRNA in OA chondrocytes reduced HA degrading activity of the chondrocytes, which expressed the other known hyaluronidases, Hyal 1 and Hyal 2 and CD44. Ding et al. and Deroyer et al. also reported the overexpression of KIAA1199 in chondrocytes of OA [21,22]. These results suggest that KIAA1199 may play a major role in degrading HA in OA chondrocytes.

Researchers in oncology identified that KIAA1199 encodes a protein that induces the migration of cancer cells and named it cell migration-inducing protein (CEMIP) [22]. Indeed, synovial proliferation has been recognized as an important event in RA progression. Deroyer et al. reported that KIAA1199 was overexpressed in human and mouse OA cartilage with dedifferentiation of chondrocytes and was essential for chondrocyte proliferation [22]. Yang et al. revealed that the survival of FLS was significantly reduced by knockdown of KIAA1199 and increased by overexpression of KIAA1199 [19], suggesting the roles of KIAA1199 in cell proliferation and survival.

Few studies have reported that it might be of help in determining whether KIAA1199, a potent hyaluronidase, can serve as a therapeutic target in inflammatory arthritis. Zhang et al. reported that serum and synovial expression levels of KIAA1199 were correlated with the amounts of low molecular weight-HA (LMW-HA) in RA patients and that administration of anti-KIAA1199 monoclonal antibody in CIA mice suppressed the severity of arthritis and reduced levels of serum LMW-HA, suggesting that the inhibition of KIAA1199-increased LMW-HA may be a possible therapeutic target in RA [23]. In a previous report, HA oligosaccharides induced a dose-dependent state of chondrocytic chondrolysis [24]. High molecular weight HA (HMW-HA) in synovial fluid normally functions as a lubricant and barrier to cytokines. On the other hand, LMW-HA promotes pro-inflammatory and pro-angiogenetic mechanisms [9,25], suggesting a correlation between LMW-HA and arthritis. In the present study, Ipriflavone was demonstrated to reduce the amount of LMW-HA and increase the amount of HMW-HA, which retained HA around the RCS cells and rebuilt the pericellular matrix. These results suggest that Ipriflavone treatment prevents HA degradation by KIA1199 and thereby interrupts the vicious cycle of OA. Ipriflavone contributed to the improvement of arthritis in CIA mice, possibly via the inhibition of stimulated HA metabolism, as shown in Figure 5. HA metabolism is upregulated in both synovia and articular chondrocytes under the influence of inflammatory cytokines [26,27]. It has been reported that the administration of Ipriflavon promotes proliferation and suppresses apoptosis of human chondrocytes [28]. Damage to the articular cartilage in RA is associated with inflammation of the synovium and cannot be treated by cartilage protection alone. Because the results of the present study that Ipriflavone downregulated the KIAA mediated HA degrading activity and suppressed MMP1 and MMP3 in FLS stimulated with TNF-α, Ipriflavone may partially contribute to the improvement of synovial inflammation and may be an option to support the treatment of RA. Ipriflavone has been used in many countries as a prescription drug and is known to inhibit bone resorption, maintain bone density, and prevent osteoporosis in postmenopausal women, although a clinical trial in 2001 reported that it was not effective in the prevention or treatment of osteoporosis [29]. In this report, 13.2% of patients in the Ipriflavone-treated group developed lymphocytopenia as an adverse event, but all patients who developed lymphocytopenia were clinically asymptomatic, and 81% of those who developed lymphocytopenia improved within 2 years. The report also stated that there were no significant differences in adverse events other than lymphocytopenia between the ipriflavone and placebo groups. As well as the treatment of monoclonal antibodies against KIAA1199, we expect that Ipriflavone will become a novel therapeutic option for treating patients with OA or RA.

There are several limitations in the present study. First, we did not measure the HA Mw of the synovium and synovial fluid of mice. The knee and ankle joints of mice were very small, making it very difficult to collect joint fluid, and it was difficult to separate the synovium when preparing joint specimens. Second, the precise mechanism that suppresses KIAA1199 activity by Ipriflavone has not been elucidated. It has been reported that the N-terminal signal sequence [30] and G8 domain [23] are important for HA degradation by KIAA1199. Ipriflavone may affect these amino acids. Further research will be needed to determine the effects of Ipriflavone on KIAA1199 with the domains such as the N-terminal signal sequence and the G8 domain.

In conclusion, we identified Ipriflavone to be a drug that suppresses the hyaluronidase activity of KIAA1199 by the drug repositioning method. Because Ipriflavone was proved to suppress inflammatory change in vitro and in vivo in the present study, and can easily be used clinically, particularly in Japan, Ipriflavone may be a promising therapeutic candidate for inflammatory arthritis via suppression of KIAA1199 activity.

## 4. Materials and Methods

### 4.1. Rat Chondrosarcoma Cells

The rat chondrosarcoma (RCS) cells used in the present study were obtained from transplantable swarm RCS with low-grade histological malignancy. They were maintained by continuing subcutaneous transplantation of the tumorous tissues in rats for several years and cultured in monolayers [31]. RCS cells have been used as a model for the metabolism of cartilage matrix molecules in vitro and in vivo [32]. No endogenous KIAA1199 mRNA expression was observed in the RCS cells (Appendix A). Mouse *KIAA1199* cDNA was transfected and stably expressed in RCS (RCS-KIAA) cells by Trap-In System, a new gene expression system combining promoter trapping and site-specific gene integration methods, as previously reported [33]. RCS cells were strongly stained around the cells with Alcian Blue, reflecting the accumulation of abundant proteoglycans, and the stainability was reduced by the overexpression of the *KIAA1199* gene (Appendix A), indicating that mouse KIAA1199 was enzymatically active.

### 4.2. Screening of 1186 FDA-Approved Compounds in RCS Cells and Ipriflavone Treatment for Cultured Cells

RCS-KIAA cells were seeded in a 96-well culture plate and incubated for 24 h in the presence of 10 μM of 1186 FDA-approved chemical compounds (Prestwick Chemical, Illkirch-Graffenstaden, France) dissolved in dimethyl sulfoxide (DMSO). After 24 h of incubation, cells were fixed with methanol for 30 min at −20 °C and stained overnight with 0.5% Alcian Blue 8 GX (Sigma, St. Louis, MO, USA) in 1 N HCl. For quantitative analyses, Alcian blue-stained cells were lysed in 6 M guanidine HCl for 6 h at room temperature [34]. The optical density of the extracted dye was measured at 490 nm and 610 nm using PowerScan 4 (DS Pharma Biomedical, Osaka, Japan). Ipriflavone (FUJIFILM Wako Pure Chemical Corporation, Osaka, Japan); dye was dissolved in DMSO to confirm its effects in indicated cells.

### 4.3. Particle Exclusion Assay

Functional cell-associated matrix was visualized and evaluated as previously described [35]. Briefly, after 24 h, culture with or without Ipriflavone in DMSO, the area without the particles around each cell was defined as the cell-associated matrix area [36] and blindly measured by Image J. The ratio of the cell-associated matrix area to the area of cytoplasmic region, which was delineated by the plasma membrane, was calculated.

### 4.4. Effects of an Identified Drug on HA Metabolism of Stimulated Synoviocytes

FLS were obtained from a patient with RA during surgery, enzymatically separated, and cultured in monolayers. Informed consent was obtained from all patients involved in the study. The cultured FLS in a tissue culture flask (1 × 10^6^ cells) were stimulated with 10 ng/mL of TNF-α for 12 h and subsequently treated with Ipriflavone (0, 2, 10 μM) in DMSO for 24 h. HA was extracted, and the Mw of each HA was determined, as previous report [32].

### 4.5. HA Quantification and Gel Filtration of Glycosaminoglycan

Solutions from RCS cells or TNF-α-stimulated FLS were subjected to HA quantification. After 24 h culture of RCS cells with or without Ipriflavone, the concentrations and Mw of HA were measured. HA was isolated according to previously described methods [32]. Briefly, the conditioned medium was collected and designated as “medium”. To extract the cell-surface-associated HA, the cells were incubated for 10 min at 37 °C with trypsin-EDTA solution and washed with phosphate-buffered saline (PBS). A mixture of trypsin-EDTA and PBS for the washes was collected and designated as “pericellular”. All samples from the cultured cells were heated at 100 °C for 15 min to inactivate protease activity, centrifuged at 15,000× *g* for 30 min at 4 °C, and then supernatants were subjected to the analyses. The HA concentrations were measured using a competitive ELISA, as described previously [37]. To determine the molecular weight of HA, HA solutions extracted from the medium of FLS cells were applied to an analytical Sephacryl S-1000 column (0.7 × 40 cm). The column was eluted with PBS solution at a flow rate of 0.5 mL/min, and each 0.5 mL fraction was collected. HA contents in the column fractions were analyzed using a competitive ELISA [37]. Mw references, including known molecular masses of HA (2130, 460, 150, and 5.6 kDa, courtesy of Seikagaku Corp., Tokyo, Japan), were used to calibrate the column.

### 4.6. Real-Time RT-PCR Analysis

Expression levels of *MMP1* and *MMP3* mRNA were determined in FLS derived from the synovial tissue of RA patients. FLS were stimulated with 10 ng/mL of TNF-α and concurrently treated with DMSO, 2, 5, and 10 μM of Ipriflavone. Total cellular RNA was isolated using RNeasy Mini Kit (Qiagen, North Rhine-Westphalia, Germany), according to the manufacturer’s instructions. Following conventional reverse transcriptase-polymerase chain reaction (RT-PCR), cDNA was subjected to real-time RT-PCR for semi-quantification of *MMP1* and *MMP3* mRNAs using a LightCycler (Roche Diagnostics, Basel, Switzerland). The relative levels of *MMP1* and *MMP3* mRNA in a sample were determined with normalization with expression levels of *GAPDH* mRNA. The *MMP1*, *MMP3,* and *GAPDH* primer pairs were as follows: *MMP1* sense; 5′-TGGACCTGGAGGAAATCTTG-3′, *MMP1* antisense; 5′-AGTTCATGAGCTGCAACACG-3′ (predicted PCR product of 135 bp), *MMP3* sense; 5′-TTCCTTGGATTGGAGGTGAC-3′, *MMP3* antisense; 5′-TGCCAGGAAAGGTTCTGAAG-3′ (predicted PCR product of 109 bp), and *GAPDH* sense; 5′-TGAACGGGAAGCTCACTGG-3′, *GAPDH* antisense; 5′-TCCACCACCCTGTTGCTGTA-3′ (predicted PCR product of 307 bp). Each mRNA level was normalized with that of *GAPDH* mRNA shown as a percentage of that of untreated control FLS.

### 4.7. Western Blot Analysis

To determine the expression of *MMP1* and *MMP3* protein, Western blot analyses were performed. Cultured FLS were lysed in the ice-cold RIPA Lysis Buffer (Santa Cruz, Dallas, TX, USA) supplemented with 0.1 mM dithiothreitol, 1 mg/mL leupeptin, 1 mM phenylmethylsulphonyl fluoride, and 1 mg/mL aprotinin. Whole cell lysates were separated on SDS-PAGE and transferred to a nitrocellulose membrane, followed by immunoblotting with anti-MMP1 (#10371-2-AP, Proteintech, Rosemont, IL, USA), anti-MMP3 (#17873-1-AP, Proteintech, Rosemont, IL, USA), and anti-ß-actin (#C4, Santa Cruz, Dallas, TX, USA) antibodies. After incubation with a horseradish-peroxidase conjugated secondary antibody (#7074, Cell Signaling, Danvers, MA, USA) and chemiluminescence reactions, images of the immunoblots were obtained using an ImageQuant LAS 4000 mini Biomolecular Imager (GE Healthcare, Chicago, IL, USA). Intensity of the immunoblots was quantified using Image J software version 1.8.0 (Wayne Rasband, Bethesda, MD, USA).

### 4.8. Collagen Induced Arthritis (CIA) Mouse Model

Seven-week-old DBA/1 J mice were purchased from Japan SLC, Inc (Shizuoka, Japan). All mice were housed in a temperature and humidity-controlled environment under 12 h light/12 h dark cycle and fed a standard rodent diet. The mice were immunized subcutaneously at the base of the tail with 100 μL of bovine type II collagen (2 mg/mL) dissolved in 0.01 M acetic acid emulsified 1:1 with Freund’s complete adjuvant (ultimately, each mouse received 100 μg of bovine type II collagen). The mice received a subcutaneously administered booster injection of 50 μg of bovine type II collagen emulsified in Freund’s incomplete adjuvant in the tail 21 days later. The study was conducted according to the guidelines of the Declaration of Helsinki and approved by the Institutional Review Board of Nagoya University Graduate School of Medicine (M210661-001, approved on 15 March 2021), and the experiments were performed according to the principles outlined in the ARRIVE guidelines.

To assess the influence of the Ipriflavone on symptom severity in the CIA model, mice were treated with Ipriflavone (100 or 200 mg/kg, *n* = 10 each) in 5% carboxymethylcellulose (CMC) solution or with CMC alone (CTL, *n* = 10) by oral feeding every day after booster immunization for 3 weeks (Appendix A). The arthritis score in these mice was scored every day and expressed as the sum of the scores of 4 limbs [38,39]. CIA mouse model was considered to have successfully developed when swelling was observed in at least 1 digit or paw, and then subsequent experiments were performed. The severity of arthritis was graded in each paw on a 0–3-point scale as follows: grade 0 = normal, grade 1 = swelling of 1 digit, grade 2 = swelling of 2 ≥ digits, grade 3 = swelling of the entire paw. The cumulative score for 4 paws of each mouse was used as the arthritis score (maximum score of 12 per mouse) to represent overall disease severity and progression in the analyzed mice [39].

### 4.9. Histologic Analysis of Knee Joints

On day 42, the mice were sacrificed under general anesthesia by systemic perfusion of 4% paraformaldehyde. The knee joints (20 each from the CTL group, 100 mg/kg, and 200 mg/kg of Ipriflavone group) were dissected and subjected to histologic analysis. The samples were fixed in 10% formalin, dehydrated, embedded in paraffin, sectioned in slides, and stained with hematoxylin and eosin (HE). Histopathologic changes in the joints were scored using the parameters described in a previous report [40]. All sections from each mouse were graded separately in a blinded manner on a 0–3-point scale as follows: grade 0 = normal, grade 1 = infiltration of inflammatory cells, grade 2 = synovial hyperplasia and pannus formation, grade 3 = bone erosion and destruction. Thus, for each mouse, the maximum possible score was 6 (sum of the score for both knee joints).

### 4.10. Serum HA Levels and Local Accumulation of HA in CIA Mouse Model

To determine the serum levels of HA, we collected blood from mouse hearts. The concentration of HA in serum was determined using an HA binding assay, as described previously [37].

The knee joint sections were analyzed histologically to detect HA using biotinylated hyaluronic acid-binding protein (b-HABP; Hokudo, Hachimen, Japan). The tissue sections were pretreated with 1 U/mL Chondroitinase ABC (pH 8.0) for 1 h at 37 °C. The tissue sections were subjected to incubation with 2.0 mg/mL b-HABP for 2 h at room temperature. Bound b-HABP was detected by the addition of streptavidin-peroxidase reagents (Nichirei, Tokyo, Japan) and diamino-benzidine-containing substrate solution (Nichirei, Tokyo, Japan).

### 4.11. Statistical Analysis

Data are presented as the mean ± SD. Statistical analyses were performed either by the Kruskal–Wallis test or one-way ANOVA followed by Tukey’s post-hoc test. *p*-values less than 0.05 were considered significant. The statistical analyses were performed using statistical database software (SPSS Statistics 27.0 software for Microsoft Windows, IBM, Chicago, IL, USA).

## Figures and Tables

**Figure 1 ijms-23-04089-f001:**
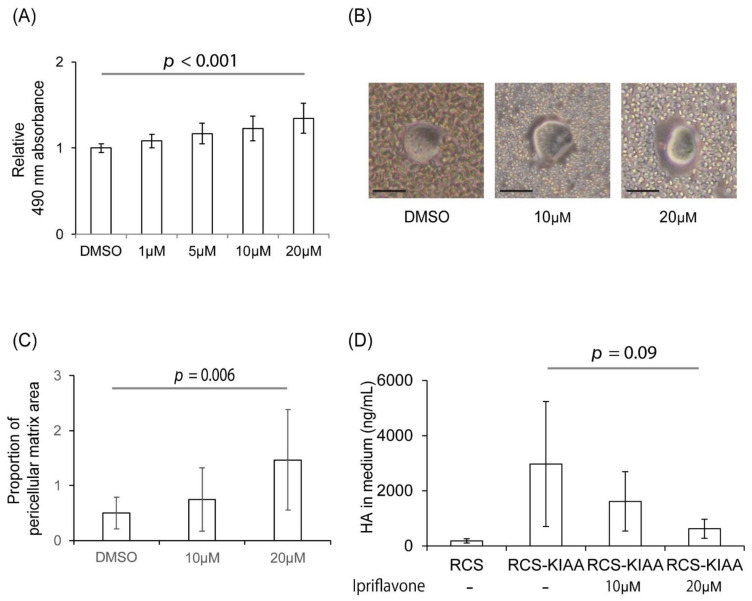
Effects of Ipriflavone on the Alcian blue staining and pericellular matrix formation of KIAA1199-transfected RCS (RCS-KIAA) cells. Mouse KIAA1199 cDNA was transfected in RCS cells. (**A**) Amount of Alcian blue-stained acidic polysaccharides such as glycosaminoglycans in RCS-KIAA cells was measured. Ratio of absorbances at 490 nm to 610 nm with or without Ipriflavone (1 μM, 5 μM, 10 μM, or 20 μM) were measured and normalized with those of DMSO-treated RCS-KIAA cells. (**B**) Representative image of RCS-KIAA cells in particle exclusion assay. Bar = 20 μm. (**C**) Ratio of pericellular matrix area compared to cytosolic area of RCS-KIAA cell with or without Ipriflavone treatment (10 μM or 20 μM). (**D**) Concentration of HA in medium of non-KIAA1199 transfected RCS cells, RCS-KIAA transfected cells with or without Ipriflavone treatment (10 μM or 20 μM). Each experiment was performed in triplicate.

**Figure 2 ijms-23-04089-f002:**
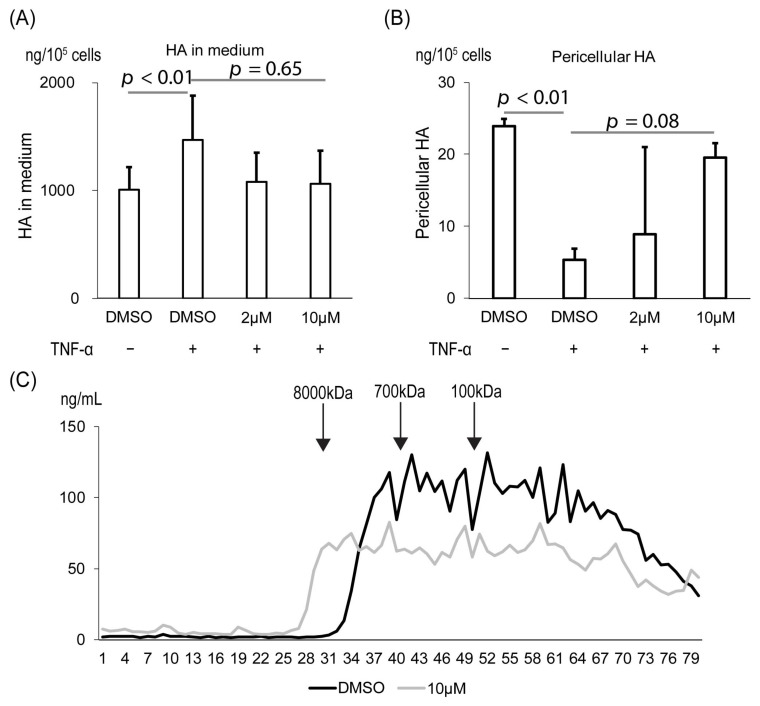
Effects of Ipriflavone on the amount of HA in cultured FLS. (**A**,**B**) FLS (10^5^ cells) was cultured with or without TNF-α and Ipriflavone (2 μM or 10 μM). Amount of HA was analyzed in the pool of conditioned medium (**A**) and pericellular area (**B**) by ELISA. (**C**) Molecular size of HA in medium from cells with or without 10 μM Ipriflavone were analyzed with gel filtration chromatography.

**Figure 3 ijms-23-04089-f003:**
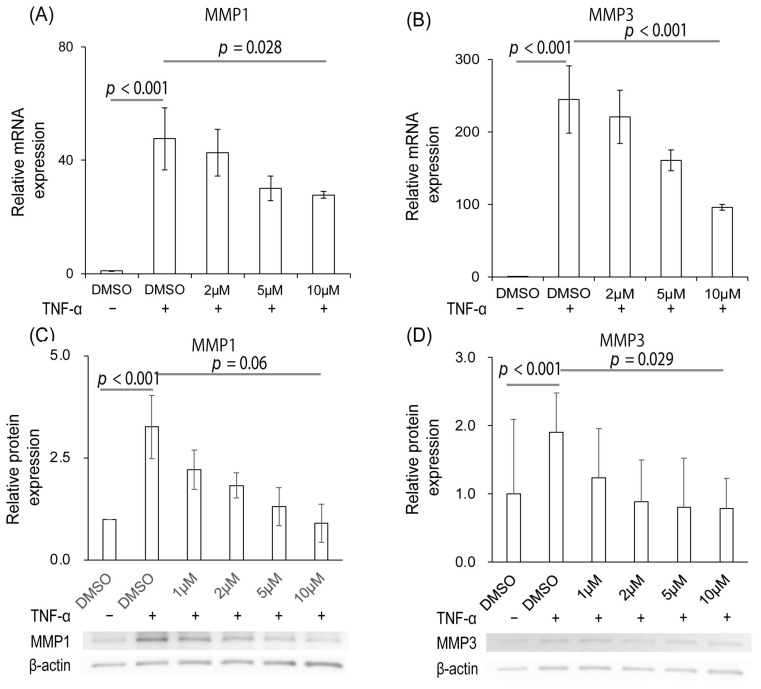
Effects of Ipriflavone on the expression of MMPs in FLS stimulated with TNF-α. (**A**,**B**) mRNA expression levels of *MMP1* (**A**) and *MMP3* (**B**) with or without TNF-α and Ipriflavone (2 μM, 5 μM, or 10 μM). (**C**,**D**) Protein expression levels of MMP1 (**C**) and MMP3 (**D**) with or without TNF-α and Ipriflavone (1 μM, 2 μM, 5 μM, or 10 μM). Experiments were repeated three times, and each experiment was performed in triplicate.

**Figure 4 ijms-23-04089-f004:**
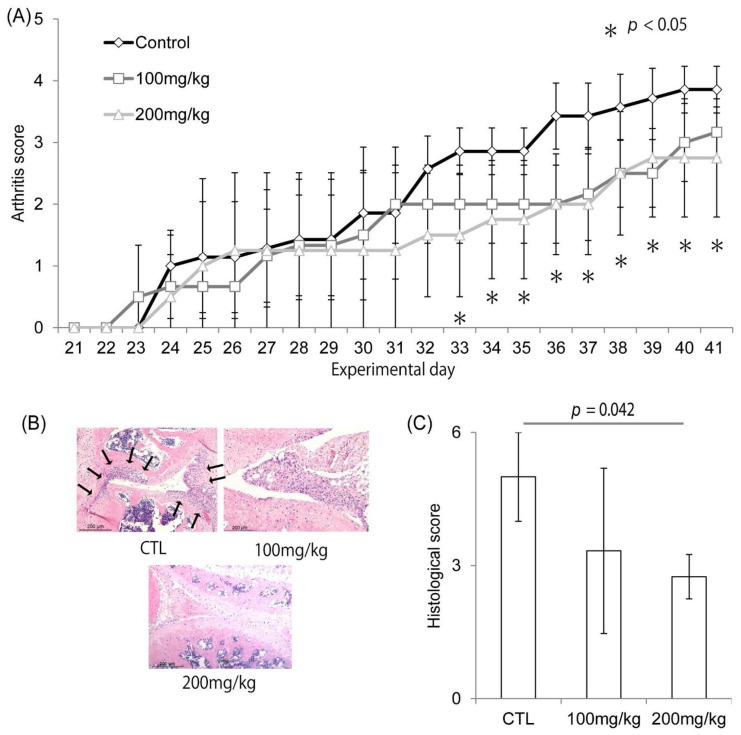
Effects of Ipriflavone on the progression of arthritis in CIA mouse model. (**A**–**C**) Effects of Ipriflavone on joint swelling. (**A**) Time course of arthritis score of paws with or without Ipriflavone treatment (CTL, 100 mg/kg, or 200 mg/kg, respectively) * *p* < 0.05. (**B**,**C**) Representative HE staining image of tissue section (**B**) and histological score (**C**) of knee joints at day 42 of CIA mice model with or without 100 mg/kg Ipriflavone treatment. Arrows depict the invading inflammatory cells in (**B**). Bar in (**B**) = 200 μm.

**Figure 5 ijms-23-04089-f005:**
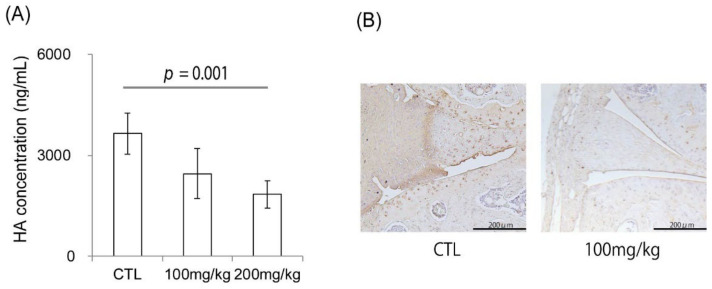
Effects of Ipriflavone on serum concentration and local accumulation of HA of CIA mouse model. (**A**) Serum concentration of HA (ng/mL) in control (CTL)-, Ipriflavone-treated mice (100 mg/kg or 200 mg/kg) on day 42. (**B**) Representative image of HABP staining for knee joints of control (CTL) and 100 mg/kg Ipriflavone treated CIA mouse model. Bar in (**B**) = 200 μm.

## Data Availability

Not applicable.

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
