# Peer review of "Possible Repositioning of an Oral Anti-Osteoporotic Drug, Ipriflavone, for Treatment of Inflammatory Arthritis via Inhibitory Activity of KIAA1199, a Novel Potent Hyaluronidase"

_ijms, 2022, doi:10.3390/ijms23084089_

Round 1

Reviewer 1 Report

The study is interesting and well designed.

However, I have some comments:
There is evidence that Ipriflavone promotes the proliferation and reduces the apoptosis of human chondrocytes, which may be beneficial in the treatment of osteoarthritis. The authors are trying to prove that Ipriflavone may become a therapeutic option in the treatment of rheumatoid arthritis. It should be emphasized that Ipriflavone can only become an option supporting this treatment, because the destruction of articular cartilage in RA is associated with the primary inflammatory process in the synovium and the treatment of the disease cannot be reduced only to the protection of the cartilage - please explain.
The authors in the discussion section describe the effectiveness of Ipriflavone in the treatment of osteoporosis. So far, researches indicate that the herbal supplement Ipriflavone is not effective at treating osteoporosis and can cause rare but serious side effects. Additionally, Ipriflavone induced lymphocytopenia in a significant number of women - please explain.

Reviewer 2 Report

This is an interesting paper. The authors  hypothesized that suppressing the enzymatic activity of KIAA1199 might inhibit the progression of inflammatory  arthritis. The purpose of this study was to search for a drug that suppresses the hyaluronidase activity of KIAA1199 by the drug repositioning method and to verify the inhibitory  effect of the identified drug on arthritis in vitro and in vivo. KIAA1199 has a strong hyaluronidase activity compared to the known  hyaluronidase and the expression of which is upregulated in the synovium of  osteoarthritis (OA) and rheumatoid  arthritis (RA)   patients.

The conclusion of the study show that Ipriflavone is a drug that suppresses the hyaluronidase activity of KIAA1199 by the drug repositioning method. Because Ipriflavone was  proved to suppress inflammatory change in vitro and in vivo in the present study, and can  be used easily clinically particularly in Japan, Ipriflavone may be a promising therapeutic  candidate for inflammatory arthritis, via suppression of KIAA1199 activity.

The introduction is complete and adequately poses the problem.

The results are clearly expressed and can be easily understood.

The discussion describes the main studies related to the use of Ipriflavone in inflammatory arthritis. The limitations of the study are expressed by proposing future research studies.

The methodology is complex and widely described and may allow the repetition of the experiments by other groups. There are supplementary files
